# Timely Monitoring of SARS-CoV-2 RNA Fragments in Wastewater Shows the Emergence of JN.1 (BA.2.86.1.1, Clade 23I) in Berlin, Germany

**DOI:** 10.3390/v16010102

**Published:** 2024-01-10

**Authors:** Alexander Bartel, José Horacio Grau, Julia Bitzegeio, Dirk Werber, Nico Linzner, Vera Schumacher, Sonja Garske, Karsten Liere, Thomas Hackenbeck, Sofia Isabell Rupp, Daniel Sagebiel, Uta Böckelmann, Martin Meixner

**Affiliations:** 1Unit for Surveillance and Epidemiology of Infectious Diseases, State Office for Health and Social Affairs (SOHSA), 10559 Berlin, Germany; julia.bitzegeio@lageso.berlin.de (J.B.); dirk.werber@lageso.berlin.de (D.W.); sonja.garske@lageso.berlin.de (S.G.); daniel.sagebiel@lageso.berlin.de (D.S.); 2amedes Medizinische Dienstleistungen GmbH, 37077 Göttingen, Germany; josehoracio.graujipoulou@amedes-group.com (J.H.G.); karsten.liere@amedes-group.com (K.L.); thomas.hackenbeck@amedes-group.com (T.H.); isabell.rupp@amedes-group.com (S.I.R.); martin.meixner@amedes-group.com (M.M.); 3Laboratory of Berliner Wasserbetriebe, Berliner Wasserbetriebe, 13629 Berlin, Germany; nico.linzner@bwb.de (N.L.); vera.schumacher@bwb.de (V.S.); uta.boeckelmann@bwb.de (U.B.)

**Keywords:** wastewater sequencing, genomic surveillance, SARS-CoV-2, Omicron, Germany

## Abstract

The importance of COVID-19 surveillance from wastewater continues to grow since case-based surveillance in the general population has been scaled back world-wide. In Berlin, Germany, quantitative and genomic wastewater monitoring for SARS-CoV-2 is performed in three wastewater treatment plants (WWTP) covering 84% of the population since December 2021. The SARS-CoV-2 Omicron sublineage JN.1 (B.2.86.1.1), was first identified from wastewater on 22 October 2023 and rapidly became the dominant sublineage. This change was accompanied by a parallel and still ongoing increase in the notification-based 7-day-hospitalization incidence of COVID-19 and COVID-19 ICU utilization, indicating increasing COVID-19 activity in the (hospital-prone) population and a higher strain on the healthcare system. In retrospect, unique mutations of JN.1 could be identified in wastewater as early as September 2023 but were of unknown relevance at the time. The timely detection of new sublineages in wastewater therefore depends on the availability of new sequences from GISAID and updates to Pango lineage definitions and Nextclade. We show that genomic wastewater surveillance provides timely public health evidence on a regional level, complementing the existing indicators.

## 1. Introduction

Monitoring the new genomic variants of severe acute respiratory syndrome coronavirus 2 (SARS-CoV-2) is crucial in addressing the public health challenge posed by COVID-19 [1]. Unfortunately, many countries have scaled down surveillance efforts, resulting in decreased PCR testing and thereby sequencing. Consequently, the number of genomic sequences of SARS-CoV-2 from patients submitted to GISAID for Germany decreased from 24,115 in October 2022 to 1401 in October 2023 and for Berlin it decreased from 944 to 220 in the same period [2]. Wastewater testing for SARS-CoV-2 has been proven useful to assess the quantitative occurrence of human SARS-CoV-2 infections [3] and the presence and distribution of SARS-CoV-2 lineages and sublineages [4,5]. Here, we present the recent emergence and dominance of the BA.2.86 sublineage JN.1, identified through specific markers in wastewater. This coincides with an increase in the hospitalization incidence and the proportion of patients in intensive care units with SARS-CoV-2 infections in Berlin.

## 2. Materials and Methods

### 2.1. Surveillance

Confirmation of SARS-CoV-2 by PCR is notifiable in Germany, with additional notification requirements for hospitals still conducting PCR tests frequently. The 7-day hospitalization rate for COVID-19 is calculated using notification data. To adjust for notification delays, the Robert Koch Institute (RKI) conducts a nowcast analysis [6]. The proportion of COVID-19 in intensive care units (ICU) in Berlin is provided by the German register of intensive care units [7]. Data on excess mortality for Berlin, Germany is provided by the European monitoring of excess mortality for public health action network (EuroMOMO) [8].

### 2.2. Wastewater Sampling and Quantification of Viral Load

Composite wastewater samples taken over a 24 h period were collected three times a week from three locations in Berlin (Appendix A), covering 84% of the Berlin population (equal to 3.25 million inhabitants). Communal wastewater was collected from raw influent by automatic samplers (WaterSam GmbH & Co. KG, Balingen, Germany). Every two hours 1 L of wastewater was collected. Afterwards, the twelve 2 h samples were mixed to one 24 h composite wastewater sample in a volume proportional manner. Using the Wizard^®^ Enviro Total Nucleic Acid Kit (A2991, Promega GmbH, Walldorf, Germany) as a direct capture method, the total nucleic acids were extracted from 40 mL of 24 h composite wastewater samples in a final volume of 50 µL following the manufacturer’s instructions. To quantify the RNA amounts of SARS-CoV-2 and of the pepper mild mottle virus (PMMoV) as a fecal indicator, digital RT-PCR analysis was performed using the QIAcuity One 5-plex dPCR system (Qiagen, Hilden, Germany), the QIAcuity 26k 24-well nanoplates (Qiagen, Hilden, Germany), the OneStep Advanced Probe Kit (Qiagen, Hilden, Germany), the GT digital SARS-CoV-2 Wastewater Surveillance Assay (GT Molecular, Fort Collins, CO, USA) and 5 µL extracted RNA as template in a 40 µL volume per reaction with three technical replicates. The data were analyzed using the QIAcuity Software Suite version 2.2.0.26 (Qiagen, Hilden, Germany).

### 2.3. Sequencing of SARS-CoV-2 RNA Fragments in Wastewater

Weekly wastewater samples, one for each of the three locations, were processed to monitor the circulating SARS-CoV-2 lineages. Concentration and purification of nucleic acids were performed as described above, followed by the illumina COVIDSeq Test, including targeted amplification with V4.1 ARTIC primer sets, library preparation and illumina short-read sequencing. Unique “fingerprint” mutations associated with COVID-19 sublineages were carefully chosen as independent markers for precise detection (see also [1]). NextStrain [9] and GISAID databases [2] provided reference for SARS-CoV-2 variants in Europe. Reads were mapped to SARS-CoV-2 via a bowtie2-samtools pipeline. Single nucleotide polymorphisms (SNP) were extracted from bam files using deepSNV R package (version 1.46.0). Sublineage prevalences are reported as determined at the time of the wastewater sampling and analysis.

### 2.4. Statistical Analysis

The trend of the PMMoV normalized SARS-CoV-2 RNA concentration in wastewater samples was modelled using a multilevel Bayesian penalized spline model [10] using rstan version 2.26.13 and brms version 2.19.0. Lineage prevalences of the three wastewater plants were averaged using a rolling mean with a tricubic weighting function and a window width of 21 days (3 weeks). All analyses were performed using R version 4.2.2.

## 3. Results

On 22 October 2023, JN.1 was detected in two of the three monitored sewage treatment plants and the concentration of the sublineage began to sharply increase until 26 November 2023 (last sampling date) (Figure 1C). This increase paralleled a decline in XBB-variants in wastewater (Figure A1).

Concomitantly to the increase in JN.1 in wastewater, the 7-day notification rate of patients hospitalized with COVID-19 infection as well as the proportion of patients in ICUs with SARS-CoV-2 infection increased steadily in Berlin until 8 December 2023 (Figure 1A,B). In sequences from clinical samples, JN.1 was first detected in Berlin on 14 November 2023 (sampled 12 October 2023).

According to the German register of intensive care units 52% (*n* = 44) of COVID-19 ICU patients in Berlin are being ventilated as of 8 December 2023. For Germany 63% of ICU patients are reported to have a COVID-19 manifestation relevant to intensive care medicine, which rises to 80% for ventilated patients. As of 21 December 2023, no excess mortality has been observed in EuroMOMO for Berlin yet (up to 6 weeks reporting delay), and the reported COVID-19 mortality for Berlin remains below 50% of the corresponding reporting weeks from last year (December 2022: 40–60 deaths per week, December 2023: 20–30 deaths per week).

Wastewater surveillance successfully identified unique mutations specific to different COVID-19 clades, ensuring an independent and focused search for key single nucleotide polymorphisms (SNPs). Furthermore, the transition from the XBB variant to JN.1 was accurately traced based on the unique mutations found in wastewater samples (Figure 2 and Figure 3, Appendix A).

## 4. Discussion

Initiated in December 2021, genomic monitoring of SARS-CoV-2 in Berlin’s wastewater promptly identified JN.1, with its sequence information available within 96 h of sampling. In contrast, patient-derived SARS-CoV-2 sequences took 2–3 weeks from sampling to submission to the national database hosted by the Robert Koch Institute (see Appendix A) [2,11]. Wastewater surveillance offers a cost-effective and timely assessment of circulating variants in the entire urban area of Berlin, based on just one sample each week per treatment plant.

JN.1 (Clade 23I) is a sublineage of BA.2.86. The latter is currently classified as variant of interest and was detected in Berlin’s wastewater at persistently low levels since July 2023. JN.1 possesses an additional spike protein mutation (S:L455S), which may enhance immune evasion capabilities [12]. The emergence of the JN.1 sublineage in Berlin coincided with an increasing trend in hospital-based COVID-19 indicators. The interpretation of these indicators has changed over time. Before the emergence of Omicron, a significant number of hospitalized COVID-19 patients, whether in regular or intensive care units, were primarily admitted due to their SARS-CoV-2 infection. However, during the dominance of Omicron, most patients infected with SARS-CoV-2 were hospitalized primarily for other reasons [13]. Consequently, these indicators, previously signaling disease severity, are now more likely to mirror the incidence in the population. This interpretation is supported for JN.1 by the observation that no corresponding (or time-delayed) increase has been registered in the mortality rate, neither among notified cases nor in EuroMOMO [8]. These hospital and mortality based parameters are complemented by our wastewater surveillance, which is our most important parameter for the general population.

While unique mutations of JN.1 could be detected as early as September 2023 in wastewater (Figure 3), these unique mutations were either classified as BA.2.86 or their relevance was unknown at the time. JN.1 could only be detected in the samples from 22 October 2023 because the new Nextclade Update for Nextstrain arrived in time for our analysis on 26 October 2023 [14]. Timely genomic wastewater surveillance depends on Pango-lineage designation updates [15] and the timely availability of many new SARS-CoV-2 sequences shared through GISAID [2].

Our results support Recommendation (EU) 2021/472 [16], which includes the tracking of variants in wastewater as “a cost effective, rapid and reliable source of information on the spread of SARS-CoV-2 in the population and that it can form a valuable part of an increased genomic and epidemiological surveillance” [16]. The genomic surveillance of wastewater for SARS-CoV-2 has been implemented in other countries and regions [1,4,17,18,19], but information from other countries is often unpublished or challenging to find. Therefore, genomic wastewater surveillance results should be collated at the EU-level to complement genomic surveillance from clinical samples.

## Figures and Tables

**Figure 1 viruses-16-00102-f001:**
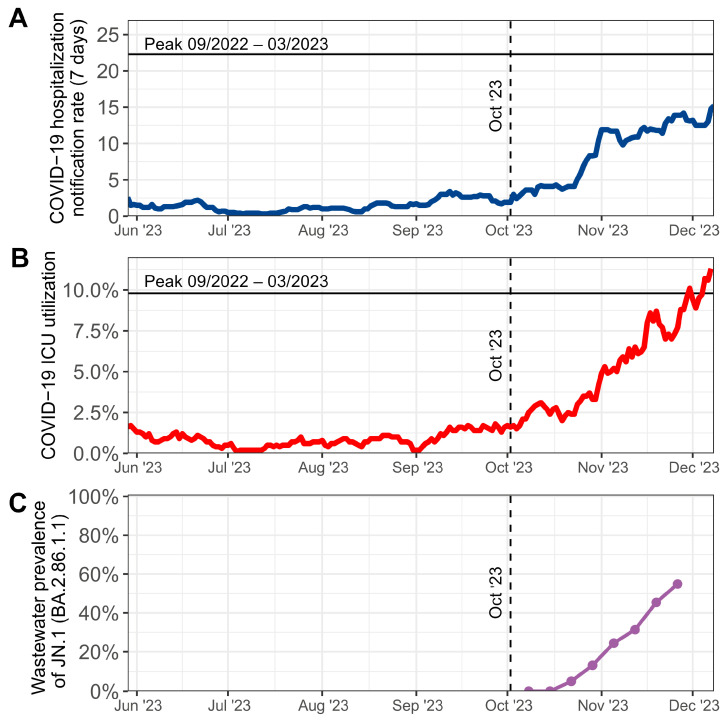
(**A**) 7-day notification rate for COVID-19 hospitalization during summer and autumn 2023 indicating an increase from October to December 2023. (**B**) COVID-19 intensive care unit utilization rate for the same time interval, also demonstrating an increase starting in October 2023. The 7-day COVID-19 hospitalization rate (**A**) and the intensive care unit utilization rate (**B**) are shown from 29 May to 8 December 2023. (**C**) Emergence of SARS-CoV-2 sublineage JN.1 (BA.2.86.1.1) during autumn 2023 in genomic wastewater monitoring for three wastewater treatment plants (Ruhleben, Schönerlinde and Wassmannsdorf), covering 84% of the Berlin population. Wastewater data is shown until 26 November 2023 (last sampling date).

**Figure 2 viruses-16-00102-f002:**
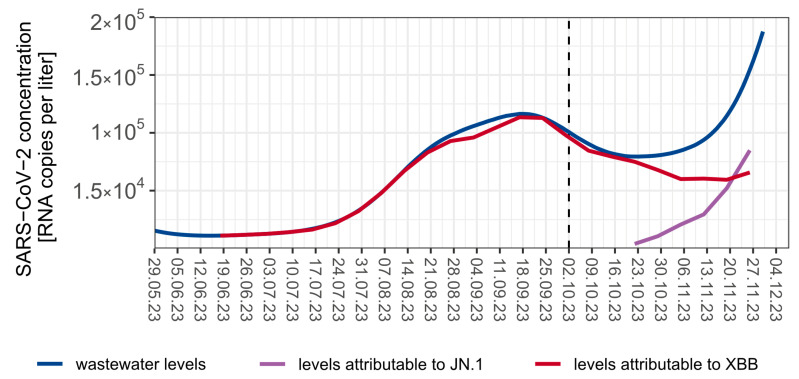
Combined SARS-CoV-2 viral concentrations in Berlin from 29 May to 26 November 2023 (last sampling date), using data from three wastewater treatment plants (Ruhleben, Schönerlinde and Wassmannsdorf), covering 84% of the Berlin population. The blue line represents the total amount of target RNA copies per liter over time, the red curve indicates XBB attributed RNA copies per liter, and the violet line shows BA.2.86.x attributed RNA copies per liter.

**Figure 3 viruses-16-00102-f003:**
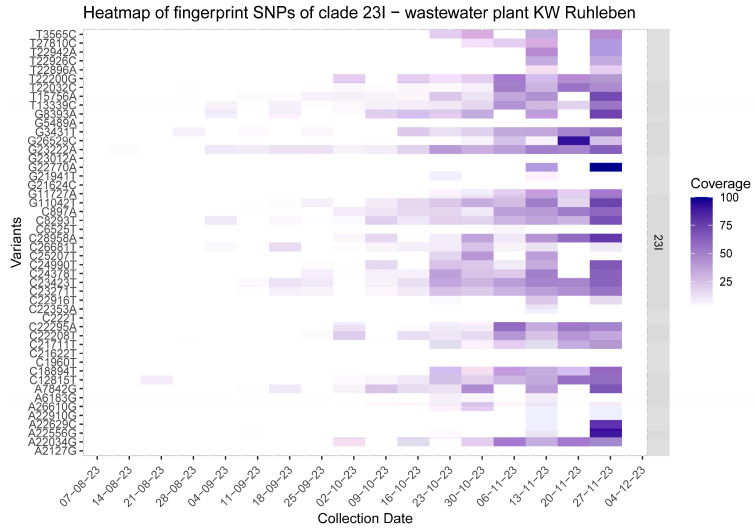
Heatmap of unique mutations found in wastewater sequencing from the Ruhleben wastewater treatment plant from 7 August to 26 November 2023 showing fingerprint variants for clade 23I. WWTP Ruhleben is shown as an example since it is the largest WWTP in Berlin covering 1.6 Mio inhabitants, the other WWTPs are shown in the Appendix A. Fingerprint mutations are exclusive for each lineage and can be used to determine the presence of a variant with a high degree of confidence. Although the fingerprint mutations for JN.1 (BA.2.86.1.1, clade 23I) were first detected in September, their significance was unknown as JN.1 would only be defined as a Pango-lineage on 29 September 2023 and was integrated in Nextstrain on 26 October 2023. Only as of the middle of October, the number and frequency of known fingerprint mutations allowed for the determination of JN.1.

## Data Availability

The COVID-19 hospitalization data for Berlin/Germany are available here [6]. Data from the German DIVI ICU registry is available here [7]. We publish wastewater monitoring data on the Berlin Open Data Portal: https://daten.berlin.de/datensaetze/COVID-19-berlin-viruslast-im-abwasser-nach-entnahmedatum-und-messstelle (accessed on 11 December 2023).

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
