# Peer review of "Timely Monitoring of SARS-CoV-2 RNA Fragments in Wastewater Shows the Emergence of JN.1 (BA.2.86.1.1, Clade 23I) in Berlin, Germany"

_viruses, 2024, doi:10.3390/v16010102_

Round 1
Reviewer 1 Report
Comments and Suggestions for Authors
This manuscript provides more support for the usefulness of wastewater sampling of SARS CoV-2 to track infection trends. Within my range of expertise, the paper appears to be of sufficient quality for publication. My comments are directed towards writing and language.
Comments on the Quality of English Language
Line 33: why are two numbers for each month mentioned? Sentence construction seems to only refer to one thing, genomic sequences submitted.
Line 55: I was taught to not start a sentence with a digit. Is this not still common practice?
Line 59; "every" two hours, not "each".
Line 61: ..volume proportional "manner"?
There are also some places where there should be another comma (or where a comma is not needed).
Author Response
This manuscript provides more support for the usefulness of wastewater sampling of SARS CoV-2 to track infection trends. Within my range of expertise, the paper appears to be of sufficient quality for publication. My comments are directed towards writing and language.
- Thank you for taking time to review our manuscript.
Line 33: why are two numbers for each month mentioned? Sentence construction seems to only refer to one thing, genomic sequences submitted.
- Wording was improved for more clarity.
Line 55: I was taught to not start a sentence with a digit. Is this not still common practice?
- Changed
Line 59; "every" two hours, not "each".
- Changed
Line 61: ..volume proportional "manner"?
- Changed
There are also some places where there should be another comma (or where a comma is not needed).
- Some additional grammatical changes were made.
Reviewer 2 Report
Comments and Suggestions for Authors
The manuscript reports a case study of wastewater surveillance on SARS-CoV-2 in Berlin, Germany. The topic is timely and the manuscript can be published after minor revision, because the reviewer noticed several points of inconsistency in the manuscript.
1) The first paragraph in the results section (On October 22nd, 2023, JN.1 ....) is not consistent with figure 1, because the data in figure 1 are plotted even after Nov. 26. In addition, the data shown in figure 1 and those in figure 2 are not consistent. The description for the y-ax in figure 1 "Wastewater proportion of JN. 1" should be more strictly defined in terms of "proportion to WHAT" (probably proportion to the total SARS-CoV-2 RNA concentration or proportion to the PMMoV concentration). The authors are required to write carefully the manuscript in order not to be misunderstood.
2) In the same way, the second paragraph in the results section (Concomitantly to the increase ...), the third paragraph (As of December 11th, 2023 ...) and the fourth paragraph (Wastewater surveillance successfully) could not be read from the figures 1, 2, and 3. The results section in a scientific manuscript on surveillance must be written based on the monitoring data. The results section must be prepared differently from the discussion section.
3) Figure 3 seems to be drawn based on the surveillance at the Ruhleben wastewater treatment plant. The reviewer did not understand why the authors focused this treatment plant out of three target plants in this study. The authors are advised to write the manuscript not to be misunderstood.
Author Response
The manuscript reports a case study of wastewater surveillance on SARS-CoV-2 in Berlin, Germany. The topic is timely and the manuscript can be published after minor revision, because the reviewer noticed several points of inconsistency in the manuscript.
- Thank you for taking time to review our manuscript.
1) The first paragraph in the results section (On October 22nd, 2023, JN.1 ....) is not consistent with figure 1, because the data in figure 1 are plotted even after Nov. 26. In addition, the data shown in figure 1 and those in figure 2 are not consistent.
- Thank you for this comment. The manuscript was updated multiple times in the days before submission, and dates were not always updated correctly. We fixed this and the date of different data sources are now more clearly stated. Wastewater data was available until November 26th 2023, Surviellance data was available until 8th December, 2023, excess mortality was last checked on 11th December, 2023 but has a large reporting delay (as stated in the paper). For more clarity we added Letters to Figure 1, and we also added figure A1 to the Appendix to show the decrease of XBB.
The description for the y-ax in figure 1 "Wastewater proportion of JN. 1" should be more strictly defined in terms of "proportion to WHAT" (probably proportion to the total SARS-CoV-2 RNA concentration or proportion to the PMMoV concentration). The authors are required to write carefully the manuscript in order not to be misunderstood.
- We updated the y-axis according to current naming standards to prevalence (i.g. Karthikeyan https://www.nature.com/articles/s41586-022-05049-6). The wastewater sublineage prevalences are estimated from snippets of mutations. Full sequences can not be easily reconstructed in wastewater, because of the multitude of different sequences present and the degradation of virus RNA.
2) In the same way, the second paragraph in the results section (Concomitantly to the increase ...), the third paragraph (As of December 11th, 2023 ...) and the fourth paragraph (Wastewater surveillance successfully) could not be read from the figures 1, 2, and 3. The results section in a scientific manuscript on surveillance must be written based on the monitoring data. The results section must be prepared differently from the discussion section.
- Data presented in the results are collected on the state level (Berlin) within the scope of our duties. This data is often collected and published on the national or European level to improve ease of access, standardization, and comparability. We therefore choose to reference these data repositories to allow easy data access for readers.
3) Figure 3 seems to be drawn based on the surveillance at the Ruhleben wastewater treatment plant. The reviewer did not understand why the authors focused this treatment plant out of three target plants in this study. The authors are advised to write the manuscript not to be misunderstood.
- The legend of Figure 3 was improved and extended. WWTP Ruhleben is the largest plant in Berlin and was therefore chosen as an example, since data was not notably different from the other WWTP. Data from all measuring sites is shown in the supplement.